# Protracted development of stick tool use skills extends into adulthood in wild western chimpanzees

Mathieu Malherbe[1,2,3]*, Liran Samuni[3,4,5], Sonja J. Ebel[6,7], Kathrin S. Kopp[6,7], Catherine Crockford[1,3], Roman M. Wittig[1,2,3]*

**1** Ape Social Mind Lab, Institut des Sciences Cognitives Marc Jeannerod, UMR5229 CNRS, Lyon, France, **2** Department of Human Behavior, Ecology and Culture, MPI for Evolutionary Anthropology, Leipzig, Germany, **3** Taï Chimpanzee Project, CSRS, Abidjan, Côte d'Ivoire, **4** Cooperative Evolution Lab, German Primate Center, Göttingen, Germany, **5** Department of Human Evolutionary Biology, Harvard University, Cambridge, Massachusetts, United States of America, **6** Comparative Cultural Psychology, MPI for Evolutionary Anthropology, Leipzig, Germany, **7** Human Biology & Primate Cognition, Institute of Biology, Leipzig University, Leipzig, Germany

* mathieu_malherbe@eva.mpg.de (MM); rwittig@isc.cnrs.fr (RMW)

**Data Availability Statement:** All relevant data are within the paper and its Supporting Information files.

## Abstract

Tool use is considered a driving force behind the evolution of brain expansion and prolonged juvenile dependency in the hominin lineage. However, it remains rare across animals, possibly due to inherent constraints related to manual dexterity and cognitive abilities. In our study, we investigated the ontogeny of tool use in chimpanzees (*Pan troglodytes*), a species known for its extensive and flexible tool use behavior. We observed 70 wild chimpanzees across all ages and analyzed 1,460 stick use events filmed in the Taï National Park, Côte d'Ivoire during the chimpanzee attempts to retrieve high-nutrient, but difficult-to-access, foods. We found that chimpanzees increasingly utilized hand grips employing more than 1 independent digit as they matured. Such hand grips emerged at the age of 2, became predominant and fully functional at the age of 6, and ubiquitous at the age of 15, enhancing task accuracy. Adults adjusted their hand grip based on the specific task at hand, favoring power grips for pounding actions and intermediate grips that combine power and precision, for others. Highly protracted development of suitable actions to acquire hidden (i.e., larvae) compared to non-hidden (i.e., nut kernel) food was evident, with adult skill levels achieved only after 15 years, suggesting a pronounced cognitive learning component to task success. The prolonged time required for cognitive assimilation compared to neuromotor control points to selection pressure favoring the retention of learning capacities into adulthood.

## Introduction

Tool use behavior, defined as the use of an environmental object to alter more efficiently the form, position, or condition of another object or organism [1,2], has been documented across various animal taxa ([2,3]; Apes: [4,5], Birds: [6,7], Insects: [8,9], Monkeys: [10]). Tool use can

**Funding:** This study was funded by the Max Planck Society (M.IF.EVAN8103 – to CC and RMW through the Evolution of Brain Connectivity Project). LS was supported by the German Research Foundation (DFG, Emmy Noether Fellowship 513871869). The funders had no role in the study design, data collection and analysis, decision to publish, or preparation of the manuscript.

**Competing interests:** The authors have declared that no competing interests exist.

facilitate access to highly nutritious food items that are otherwise inaccessible, as well as fulfill other functions such as communicating or gathering sensory information [11]. It has been suggested that tool use behavior is differentiated into *stereotyped* and *flexible* use [12]. Whereas stereotyped tool use necessitates individual learning but not reasoning, *flexible* tool use requires social input, individual reasoning, and understanding of action causality [12,13]. Flexible tool use is particularly relevant during tasks in which multiple steps are required to be successful or when the perceptual distance between the problem and solution is large (e.g., extractive tool use wherein the target is not directly visible) [14,15]. The way in which humans use tools flexibly is unmatched [16]. One hypothesis that explains this ability in humans is the capacity to continue to learn across the entire lifespan, which is considered connected to the human phenomenon of having both prolonged brain maturation and prolonged juvenile dependency [17]. While other species also show prolonged development and parental association (e.g., apes, whales, elephants, New Caledonian crows), it remains unclear if this relates to prolonged learning of complex foraging, here examined in tool use behavior.

For many species, the capacity to manipulate a tool with precision, coordination, and control is physically constrained by morphological features. For example, New Caledonian Crows compared to other avian species have a bill morphology adapted for enhanced tool manipulation [18,19], and primates have opposable thumbs which offer improved motor dexterity. Thumbs have been hypothesized to be essential in facilitating progression in tool use and tool making during human evolution [20–24]. The ability of humans to independently oppose their thumb to their other digits offers the possibility to manipulate objects and tools with either power and/or precision [23]. However, not all primates' thumbs are equivalent, with intrinsic hand proportions (e.g., how long the thumb is relative to the other digits, how many muscles are involved, what is the mobility at the thumb to wrist joint) likely limiting the manipulative abilities of different species. Examining the use of the thumb and other digits during tool use behavior of extant nonhuman primates can offer a comparative perspective on the evolution of morphology related to tool use and assess the extent to which morphology constrains tool use skills.

Alongside morphology, cognition is also considered as a potential constraint on tool use behavior and development (see [25]). For example, flexible tool use behavior is thought to require the ability to combine different experiences to solve novel problems, otherwise known as reasoning [13,26], conceptual knowledge about objects [27], working memory [14], hierarchical cognition [28], and socio-cognitive capacities such as emulation and imitation in the case of socially learned tool use [29]. Comparative studies of nonhuman animals show considerable variation in relevant cognitive capacities (e.g., [30–32]), highlighting the potential for cognition to explain why flexible tool use is observed in some species but not others.

While a number of different species have been shown to use tools flexibly (e.g., gorillas [33], sea otters [34], orangutans [35], New Caledonian crows [6], bottlenose dolphins [36], Goffin's cockatoo [37], woodpecker finch [38], and bearded capuchins [39]), chimpanzees are one of few species where flexible tool use behavior is regularly observed across individuals, populations, tool materials, and contexts [40]. As one of our closest living relatives, chimpanzees share some similarities with humans that may inform tool use acquisition and use. The ability to socially learn from others allows for material traditions and culture to exist between communities and populations [40,41]. While significant differences exist between the hand anatomy of chimpanzees and humans, with the former having long digits and small and weak thumb considered inoperant in precise handling [22], chimpanzees employ all hand grips described in humans [23,42–46], including the pad-to-pad precision grip, a complex hand grip long thought to be unique to humans ([47,48]; although see: [22,24,46]).

Chimpanzees also share with humans a prolonged developmental period and maternal dependency [49–53]. Within the hominin lineage, it has been hypothesized that prolonged juvenile dependency, which is related to parental provisioning, facilitated prolonged brain development (e.g., [54]), which in turn enabled protracted learning capacities needed for complex foraging and tool use [17]. Chimpanzees may not continue extensive parental provisioning after weaning at age 5 years old, however, individuals that experience maternal loss after weaning and before adulthood compared to those that do not, lose out on fitness [50,52,53]. This suggests that mothers continue to offer benefits to offspring, although what these benefits are, is not well understood. One possibility is that acquiring the skills for the highly flexible tool use demonstrated by chimpanzees requires protracted learning across development. Specifically, we predict that tool use skills to extract difficult-to-access, high-nutrient foods, may require protracted learning. To assess this, we tease apart development of the motor control needed to manipulate tools (hand grip type) and skill acquisition required to fit specific tool actions to food tasks.

Previous studies have proven instrumental for our understanding of the processes involved in chimpanzee tool use acquisition [55–57]. However, due to their slow life histories, studies examining how chimpanzee tool use manipulative skills develop across their lifetime, especially in the wild, are rare. This gap in the literature impedes our understanding of the motor and cognitive skills that may underlie chimpanzee natural tool use behavior. Here, we capitalized on a video dataset of 70 wild chimpanzees using sticks as tools for extractive foraging, collected over 7.5 years. Using cross-sectional data, we examined how the hand grip and action used to manipulate stick tools varied across ontogeny and tool use tasks. We hypothesized that ontogeny will lead to the maturation of both the grip to hold and use a stick tool, and the task understanding in the context of stick tool use. To assess hand grip use, we examined (i) how hand grip preferences change with age; and (ii) whether grips involving the use of one or several digits independently increased accuracy and efficiency. To assess skill specialization and task understanding, we examined (i) whether hand grips changed according to the action required to complete the task (i.e., whether requiring power or precision); and (ii) whether action selection per task became more specific with age. If learning is involved in these choices, we expected older individuals to be better at using the suitable hand grip and actions for a given task compared to younger individuals.

Given the protracted developmental dependency observed in chimpanzees, we expect that chimpanzee tool manipulation skills will develop slowly through ontogeny, extending beyond weaning age. Specifically, as motor control matures, we expect stick tool hand grip to become optimized to the task in hand. When task experience is required, we expect learning the action required to extract difficult to access foods using stick tools will continue to develop for some years after hand grip motor control has become proficient.

## Materials and methods

### (a) Ethics statement

All methods were non-invasive and were approved by the Ministère de la Recherche et de l'Environnement of Côte d'Ivoire and also by the Office Ivorien des Parcs et Réserves. All aspects of the study comply with the Department of Primatology ethics policy of the Max Planck Institute for Evolutionary Anthropology (Ethikrat der MPG 04/08/2014).

### (b) Study site, subjects, and data collection

This study used video data of stick tool use collected on 3 neighboring communities (i.e., North group, East group, and South group) of wild western chimpanzees (*Pan troglodytes*

*verus*) from November 2013 to April 2020 in the Taï National Park in Côte d'Ivoire, West Africa (5˚45N, 7˚7W) [58]. During the study period, the group sizes varied between 29 and 34 individuals in the East group, 37 and 41 individuals in the South group, and 20 and 24 individuals in the North group. The video data were collected ad libitum while performing all-day focal animal sampling of adult individuals from the different communities [59]. From this ad libitum Taï chimpanzees video library, MM coded 135 videos (i.e., 15,154 s) using BORIS software [60], depicting stick use events extracting food from any cavity the stick needed to be inserted into, where the tool, hand, and the food sources were visible. From these videos, an overall of 1,460 stick use events (see definition section (d) below) of 70 chimpanzees (139.67 ± 152.60s) ranging from age 1 to 54 years old were extracted (see Supporting information S1–S4 Tables for a thorough description of the dataset). Inter-observer reliability tests, where each observer coded 10% of the video recorded time, were performed between MM and 3 different observers which resulted in strong agreement (i.e., *grip coding*: Cohen's Kappa observer 1: 0.811, Cohen's Kappa observer 2: 0.831; *action coding*: Cohen's Kappa observer 3: 0.826).

## (c) The different types of hand grip observed in Taï

Hand grips in different species have been named differently, even when using the same digits in the same way (see S8 Table in the Supporting information material for a full review of all grips presented in this section and the different names given to them in other studies with the studied species associated). We named the grips after what we think is important in holding and guiding the tool, thereby the different hand grip types (Fig 1) observed in Taï were defined following the number of digits used by the chimpanzees:

i. The *full hand grip* was defined by the stick tool being held in a clamp formed by 4 flexed digits, with or without the palm being involved. The thumb was either not involved or placed in adduction on the other digits. No independent digit was used to guide the stick tool. This grip is also called the "full hand wrap grasp" (see #1, #2, #3 in Feix and colleagues [61]) or the "Transverse hook" [62,63].

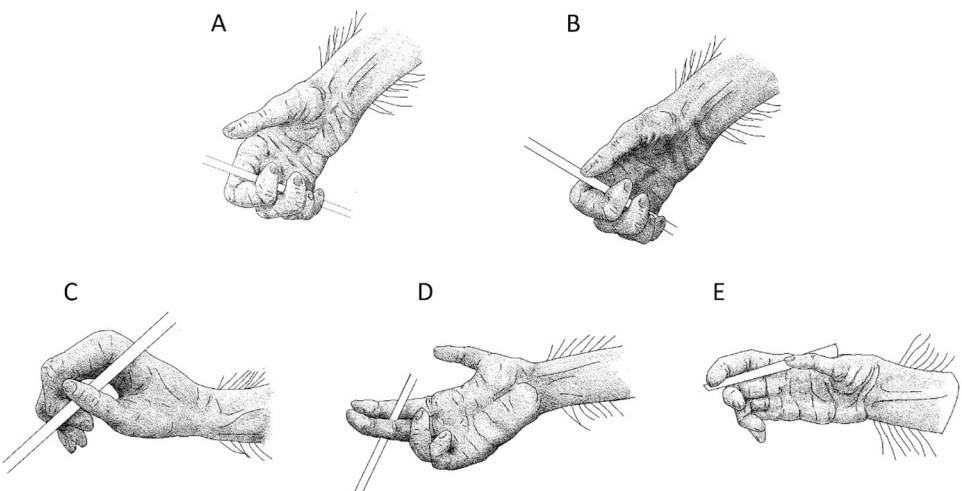

**Fig 1.** Representation of the different hand grip used by chimpanzees of the Taï National Park in order to hold and manipulate a stick tool to access food: (A) *full hand grip*, (B) *full hand thumb grip*, (C) *digits grip* category 1, (D) *digits grip* category 2, (E) *digits grip* category 3. Drawn by Oscar Nodé-Langlois, TCP.

ii. The *full hand thumb grip* was defined by the stick tool being held in a clamp formed by 4 flexed digits, with or without the palm being involved. The thumb in extension being the only digit used independently to guide the stick tool. This grip is also called the "Light tool" grip (see #5, [61]).

iii. The *digits grip* was defined as using 2 or 3 independent digits to hold and guide the stick during the tool use behavior. Within the *digits grip* type, several hand grips were observed and classified into 3 categories (Fig 1):

  a. Category 1: *digits grip* with the stick tool being held between the thumb and the side of the index. The stick could be oriented either to the palm or to the back of the individual's hand. This grip is also called the "lateral grip" (see #16, [61]) or the "two-jaw chuck pad-to-side" [62,63].

  b. Category 2: *digits grip* with the stick tool being held between the side of 2 different digits, without the thumb being involved. The stick could be oriented either to the palm or to the back of the individual's hand. This grip is also called the "adduction grip" (#23, [61]) or the "scissor hold" grip [62,63].

  c. Category 3: *digits grip* with the stick tool being held between the side of 2 digits with the thumb being involved as a third digit. This grip is also called the "tripod variation" (#21, [61]).

## (d) Video coding

All videos were coded using the open-source Behavioral Observation Research Interactive Software (BORIS; [60]). For each stick tool use behavior, we identified the subject using the tool and coded the following variables:

Event: Defined as a stick tool use behavior from an individual that starts from the first contact of the stick to the surrounding of the hole with the food source to the moment the individual investigates the stick after removing it from the hole, or discards the stick.

Context: Defined as the type of resource the stick tool was directed towards (i.e., the resource individuals attempted to extract: honey, ants, larvae, termites' in or on the mound, bone marrow, beetle, nut kernel, or seeds extracted from pods—example videos are available on www.taichimps.org). Precision, power, and dexterity needed per task depends both on the hole size and nature of resource.

Session: Defined as a sequence of stick tool use events taking place within a specific location. Recording the sessions helped account for site-specific physical characteristics that may impact tool use behavior.

Action: Defined as the different behavior chimpanzees used to access and process the food resources (see Table 1).

Hand grip type: Defined as the type of hand grip described in *section c*, coded for each action within a stick tool use event from the different videos.

Number of attempts: Defined as the number of trials individuals need to insert the stick into the hole.

Food amount: Defined as the amount of food extracted during stick tool use (only objectively feasible in beetle and larvae extraction), coded in 3 categories:

i. None: no food retrieved during the stick use event.

ii. Partial: pieces of a squashed food retrieved during the stick use event.

iii. Whole: entire intact food retrieved during the stick use event.

**Table 1. Description of the different actions performed with a stick tool by 3 communities of habituated wild western chimpanzees (*Pan troglodytes verus*) in the Taï National Park, Côte d'Ivoire.** The different levels of power and precision are represented for the different actions (+: high, ±: intermediate, -: low).

| Actions | Description | Power | Precision |
|---|---|---|---|
| **Insert** | The stick is entered into the cavity. | - | + |
| **Lever** | The stick is used like a spoon to extract the food source from the cavity. | ± | ± |
| **Prod** | The stick pokes the inside of a cavity in a movement aligned with the direction of the hole. | ± | ± |
| **Screw** | The stick is rotated in the cavity in a radius no larger than the stick itself. The stick rotated between the fingers or fixed in the hand while the wrist rotates. | ± | ± |
| **Stir** | The stick is rotated in the cavity in a radius larger than the stick itself. | ± | ± |
| **Pound** | The stick is used to hit the food source in a movement aligned with the direction of the hole. | + | - |

## (e) Statistical analyses

To test our hypotheses, we fitted Bayesian Regression models in R (version 4.1.2) using the R package "brms" [64]. The model predictions and structures are presented in Table 2. In order

**Table 2. Table presenting the models predictions and structures.**

| Model 1 –The ontogeny of stick tool hand grips (categorical model) | |
|---|---|
| Prediction | The hand grip with no independent use of digits (i.e., *full hand grip*) will be used by younger nonexpert individuals (i.e., infants), who will then switch to the hand grip using 1 independent digit (i.e., *full hand thumb grip*) after some experience (juveniles), while expert (adult) tool users will use a hand grip with 2 or 3 independent digits to hold and guide the stick tools (i.e., *digits grip*). |
| Model formula | Hand grip ~ Age + Age$^2$ + Sex+ Group + (1+Age+Age$^2$\|Context) + (1\|Subject ID) + (1\|Session) |
| **Model 2 –Hand grip accuracy (Poisson model)** | |
| Prediction | Hand grips involving more independent use of digits will be more accurate. Young and old individuals will exhibit less motor control and, therefore, may require more attempts. |
| Model formula | Number of attempts ~ Hand grip + Age + Age$^2$ + Sex+ Group + (1+Age+Age$^2$\|Context) + (1\|Subject ID) + (1\|Session) |
| **Models 3A and 3B –Hand grip suitability for performing required action–across ages (categorical model)** | |
| Prediction | Adult (>15 years) individuals will use the more adequate hand grip to perform the action (model 3A). No such modulation of behavior would be observed for non-adults (<15 years) individuals (model 3B). |
| Model formula | Hand grip ~ Action + Sex + Group + (1 \|Context) + (1\|Subject ID) + (1\|Session) |
| **Model 4 –Action suitability for food retrieval (categorical model)** | |
| Prediction | Levering will be more efficient in extracting larvae than other actions, and adults (>15 years) compared to immatures will extract larger amounts of larvae. |
| Model formula | Amount of larvae ~ Levering action (y/n) + Age class + Sex + (1\|Subject ID) + (1\|Session) |
| **Model 5 –Likelihood to choose appropriate action with age (Bernoulli model)** | |
| Prediction | Adult individuals (>15 years) will be more likely to use the levering action in the larvae extraction context than non-adult (<15 years) individuals. |
| Model formula | Levering action (y/n) ~ Age class + Sex + (1\|Subject ID) + (1\|Session) |
| **Model 6A and 6B –Action choice in 2 different food tasks through ontogeny (Bernoulli model)** | |
| Prediction | Use of levering will be reached later in ontogeny for hidden foods (larvae extraction—model 6A) than for visible foods (nut kernel extraction -model 6B) |
| Model formula | Levering action (y/n) ~ Age + (1\|Subject ID) + (1\|Session) |

to keep type 1 error rates at the nominal level of 5%, we included the random slopes within the adequate random effects variables, as well as the correlation parameters between the random intercepts and random slopes terms [65,66].

Moreover, in all the models, we z-transformed continuous predictors to a mean of zero and a standard deviation of one. We ran 2,000 iterations with a "warm-up" of 1,000 iterations over 4 MCMC chains, leading to 4,000 posterior samples [67]. Rhat values inspection of all MCMC revealed satisfactory values (<1.01; [68]). No divergent transitions after warm-up and stationary and convergence to a common target were visually observed, suggesting our results are stable. In addition, our models did not suffer from collinearity issues which we evaluated using variance inflation factors [69] with R package "car" [70]. For all models, we report hereafter the estimate (mean of the posterior distribution) and the credible intervals (CIs) at 2 different levels: CI 89% and CI 95% [71].

**Model 1: Development of hand grip.**   To examine the development of hand grip types, we examined how hand grip decisions varied by individuals' age. We focused on the insertion task, the first action to reach and consume the hidden food source. Linear and squared terms of age were included as fixed effect variables as we predicted a nonlinear effect of age on the hand grip type used (see prediction model 1 in Table 2). The dataset included 135 videos of 70 individuals ranging from 1 to 54 years of age, with 1,460 stick-use events coded across the seven-and-a-half years of data collection. From this dataset, 25 individuals were observed across different ages (i.e., between 2 and 6 different ages).

**Model 2: Hand grip accuracy.**   To explore hand grips' impact on the accuracy of accessing food sources with a stick tool, we tested whether the number of attempts was influenced by the hand grip type used to insert stick tools into holes. We also tested the effect of age on the accuracy measure where we expected age to have a nonlinear effect on the number of attempts to reach the food sources (see prediction model 2 in Table 2), hence we included both linear and squared terms of age in our model. We used 1,386 observations of 68 individuals (aged 1 to 54) for this model. Stick use events where we could not account for the number of attempts (5%) were excluded from this analysis.

**Model 3: Hand grip used in power and precision actions.**   To test task understanding by adult and non-adult chimpanzee tool users in Taï when manipulating a stick, we examined how chimpanzees flexibly use different degrees of precision and power in their tool grip. We used 2 actions representing the 2 extremes of required power versus precision in our dataset: (i) the inserting action that we predicted would require the most precision with no element of power; and (ii) the pounding action that we predicted would require the most power (see section d, Table 1). The other actions were excluded from this analysis as they would have been difficult to classify objectively concerning the interplay between precision and power. We built 2 separate models, the first where we included only adult (i.e., individuals of 15 years old or older) tool users, which are considered experts in tool using (model 3A) and the second where we included only non-adult (individuals younger than 15 years) tool users which are considered not experts yet (model 3B). For the "adult" and "non-adult" model respectively, a total of 827 and 152 different stick use events were accounted for in our analysis which were performed by 12 and 8 different individuals aged between 15 and 54 or 3 and 14 years old, respectively. Individuals that did not perform any pounding action through the entire dataset were excluded from this analysis.

**Model 4: The use of the adequate action to extract high nutrient food.**   To test whether different actions facilitated the extraction of more food, we examined whether individuals retrieved more food when using the levering action in the larvae extraction context. We chose the larvae extraction context as it was one of the only contexts where the amount of food could be objectively assessed. We considered the levering action as it represents the most appropriate

action to displace and retrieve higher amounts of larvae with a stick tool from a narrow cavity. All but 2 of our observations of larvae extraction took place in only 1 group (i.e., North group), hence why we decided to exclude the group identity variable from our analysis. In total, 137 actions were accounted for in our analysis, which were performed by 12 different individuals aged between 4 and 43 years old.

**Model 5: Probability to use the adequate action at different age.**   If the result of model 4 confirmed that the use of the levering action improves food extraction, we additionally tested if the use of levering differed between adult and non-adult individuals. In such a case, we used the same dataset as in model 4.

**Model 6: Development of the adequate action in different contexts.**   To model the developmental trajectory of actions, we examined the use of the levering action in 2 different contexts (i.e., the nut kernel extraction and the larvae extraction). As we were interested in examining the transition of action use into adulthood, we included in this model only individuals between 1 and 20 years of age. We argue that the use of the levering action is less complex for nut kernel retrieval compared to larvae extraction due to the food characteristics. In the nut kernel retrieval, the levering action targets broken pieces of visible nut that remained in the shell after the shell was cracked open. In the larvae extraction context, the levering action aims to bring a living larvae hidden from view to the hole's entrance without crushing it. We fitted 2 models to examine this question, one for each extractive foraging context. In the larvae extraction model, our observations were from North group and we found a collinearity issue between the age and sex of the individuals. In order to have consistency, we decided to exclude both the group identity and sex variables from the larvae extraction and nut kernel extractions models. For the larvae extraction and nut kernel extraction model respectively, a total of 110 and 65 different actions were accounted for in our analysis which were performed by 8 and 6 different individuals aged between 4 and 19 or 2 and 18, respectively.

## Results

### Development of hand grips

We identified categories of hand grips. Each was observed in all age-classes (Table 3). Chimpanzees younger than 5 favored the *full hand thumb grip* ($N = 59$ (57%) of 103 stick tool use events, compared to $N = 14$ and $N = 30$ for *full hand grip* and *digits grip*, respectively). In contrast, *digits grip* was the most observed (>61%) hand grip used in the other age-classes. With increasing age, chimpanzees used the *full hand thumb grip* less often (age-class 5–9: 38%; age-class 10–14: 31%; age-class 15–19: 10%) and the *digits grip* more often (age-class 5–9: 61%; age-class 10–14: 69%; age-class 15–19: 89%).

We tested if and how age affected the probability of using a particular grip type to insert a stick tool into a hole for extractive foraging (Fig 2 and Table 4). We found that both the linear

**Table 3. Total number of hand grip types used during stick tool use (and number of videos in parentheses), by age class, observed across habituated wild western chimpanzees communities (*Pan troglodytes verus*) in the Taï National Park, Côte d'Ivoire.**

| Age-class (in years) | 1–2 | 3–4 | 5–9 | 10–14 | 15–19 | 20–29 | 30–39 | 40–49 | 50–54 |
|---|---|---|---|---|---|---|---|---|---|
| Number of individuals | 10 | 10 | 14 | 9 | 17 | 13 | 4 | 3 | 1 |
| Full hand grip | 10 (7) | 4 (3) | 2 (2) | - | 2 (2) | 4 (2) | 6 (2) | 1 (1) | - |
| Full hand thumb grip | 13 (4) | 46 (11) | 92 (19) | 31 (9) | 45 (19) | 13 (6) | 1 (1) | 2 (1) | - |
| Digits grip | 4 (4) | 26 (10) | 145 (26) | 69 (19) | 399 (46) | 297 (37) | 55 (10) | 119 (16) | 74 (6) |
| Total | **27 (15)** | **76 (24)** | **239 (47)** | **100 (28)** | **446 (67)** | **314 (45)** | **62 (13)** | **122 (18)** | **74 (6)** |

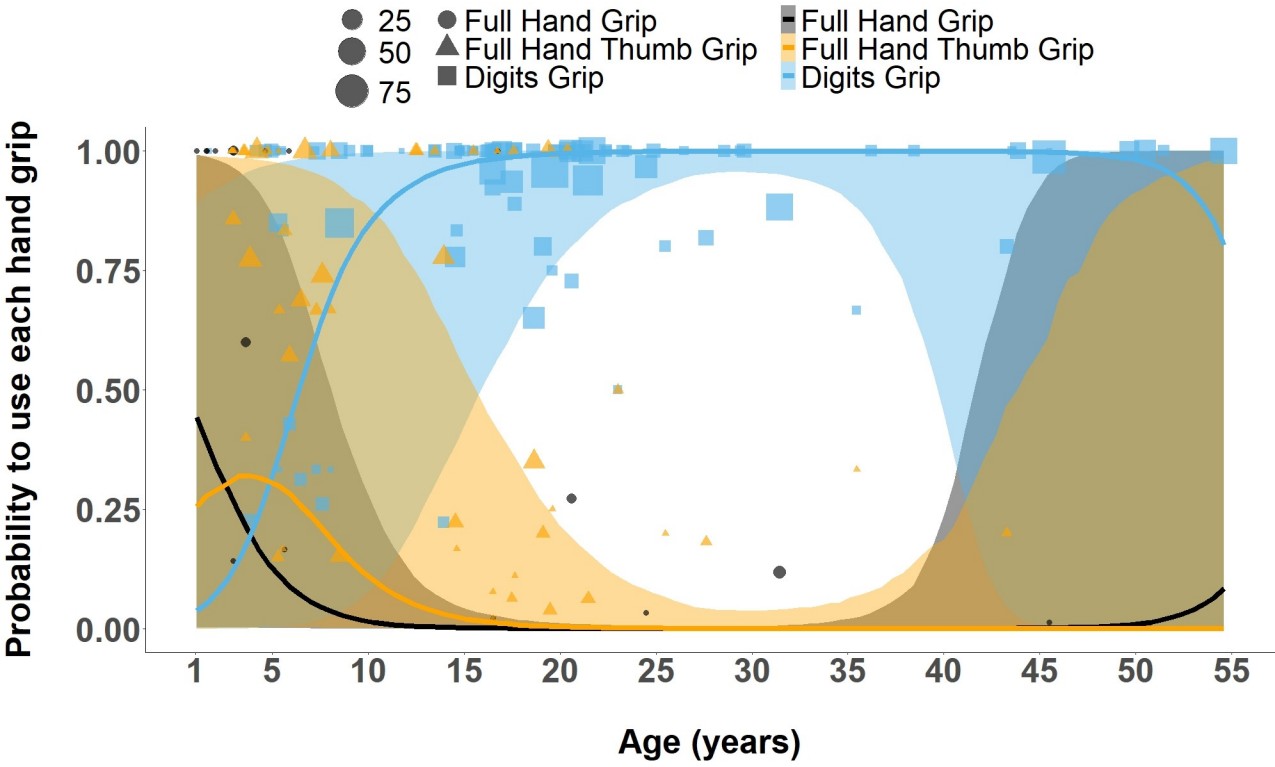

**Fig 2. Effect of age on the probability to use different hand grip types (*full hand grip* = black, *full hand thumb grip* = orange, and *digits grip* = blue) during stick tool use events.** The regression lines from the Bayesian regression model show a 95% CI for the 3 hand grip types. Black circles (*full hand grip*), orange triangles (*full hand thumb grip*), and blue squares (*digits grip*) sizes represent the number of hand grips coded per individual per age. The data and code needed to generate this figure can be found in S1 Data and S1 Code.

(estimate: 3.50, CI 95%: [1.07, 6.04]) and the squared (estimate: −2.36, CI 95%: [−5.04, −0.41]) terms of age had a strong effect on the probability to use the *digits grip* (with *full hand grip* as the reference category). The model predicted that the probability of using the *digits grip* was 5% (response probability shown in Fig 2: 0.054) at the age of 1 year and slowly increased to become the predominant hand grip used (37%, response probability shown in Fig 2: 0.368) by 5.18 years old. The *digits grip* usage continued to increase throughout the immature years, reaching 98% (response probability shown in Fig 2: 0.978) probability to be used at 15 years old. The probability of using the *digits grip* started decreasing at 36 years old (model prediction: 36.67y), reaching an 80% likelihood at 54 years old (response probability shown in Fig 2: 0.80; model prediction: 54.58y). However, only 3 females from 1 group contributed to the data for ages above 45 years.

We also found a moderate but non-robust (i.e., <89% CI) effect of age² on *full hand thumb grip* (estimate: −1.78, CI 84%: [−3.69, −0.03]). By the age of 2 years old (model prediction: 2.46), *full hand thumb grip* became the predominant hand grip type in the population, with its highest probability of being used (response probability shown in Fig 2: 0.32) observed at the age of 3 years (model prediction: 3.55y). Afterward, the probability of using *full hand thumb grip* declined again, being taken over as the predominant grip type in the population by *digits grip* at 5 years old and nearly disappearing by 19 years of age (model prediction: 19.84y; response probability shown in Fig 2: 0.0043).

In comparison, *full hand grip* is the predominant hand grip type used at 1 year old (response probability shown in Fig 2: 0.38) but decreases to a probability close to 0 by the age

**Table 4. The effect of age, sex, and group identity on hand grip type used to insert stick during stick tool use events.** Credible intervals of 89% and 95% are presented. All numeric predictor variables were standardized to mean = 0 and sd = 1. Values in bold represent credible intervals excluding zero.

| Coded level | Term | Estimate | SE | 89% CI | 95% CI |
|---|---|---|---|---|---|
| *Full hand thumb grip*[a] | Intercept | 2.53 | 2.22 | −0.89, 6.21 | −1.71, 7.16 |
| | Age | −0.54 | 1.45 | −2.87, 1.65 | −3.60, 2.23 |
| | Age$^2$ | −1.78 | 1.38 | −4.03, 0.28 | −4.62, 0.84 |
| | Sex[b] | 2.07 | 1.93 | −0.93, 5.19 | −1.60, 5.99 |
| | Group North[b] | 4.48 | 3.50 | −0.83, 10.32 | −2.09, 11.85 |
| | Group South[b] | −2.80 | 2.09 | −6.32, 0.40 | −7.19, 0.92 |
| *Digits grip*[a] | Intercept | 8.25 | 2.13 | 5.09, 11.77 | 4.45, 12.90 |
| | **Age** | 3.50 | 1.25 | **1.69, 5.49** | **1.07, 6.04** |
| | **Age$^2$** | −2.36 | 1.18 | **−4.30, -0.82** | **−5.04, -0.41** |
| | Sex[b] | 2.47 | 1.63 | **0.03, 5.20** | −0.48, 5.92 |
| | Group North[c] | 4.13 | 3.41 | −0.81, 9.79 | −1.95, 11.70 |
| | Group South[c] | 0.48 | 1.79 | −2.32, 3.38 | −3.03, 4.07 |

Reference categories for the variables presented in this table are as follow:

(a): *Full hand grip*

(b): Female

(c): Group East

of 12 years (response probability shown in Fig 2: 0.043, model prediction: 12.78y). The probability of *full hand grip* increases again around the age of 30 years (model prediction: 30.7y) and reaches an 8% (response probability shown in Fig 2: 0.084) probability of being used at 54 years old (model prediction: 54.58). An effect of the control predictor of individual's sex at an 89% credible interval was observed (estimate: 5.35, CI89%: [0.03, 5.20]), with males using a higher proportion of *digits grip* than females.

## Accuracy of stick insertion depending on hand grip type used

We tested if the hand grip type used impacted the accuracy of the tool use behavior using the number of attempts to insert a stick into a cavity during natural foraging tasks as a proxy for

**Table 5. The effect of the hand grip type used, the age of the individual performing the task, its sex, and its group identity on the number of attempts to insert stick tools into high nutrients food holes.** Credible intervals of 89% and 95% are presented. All numeric predictor variables were standardized to mean = 0 and sd = 1. Values in bold represent credible intervals excluding zero.

| Term | Estimate | SE | 89% CI | 95% CI |
|---|---|---|---|---|
| Intercept | 0.78 | 0.21 | 0.45, 1.10 | 0.37, 1.19 |
| *Full hand thumb grip*[a] | −0.46 | 0.15 | **−0.70, −0.22** | **−0.75, −0.16** |
| *Digits grip*[a] | −0.36 | 0.14 | **−0.58, −0.14** | **−0.63, −0.07** |
| Age | −0.10 | 0.08 | −0.21, 0.02 | −0.25, 0.06 |
| Age$^2$ | 0.05 | 0.06 | −0.03, 0.14 | −0.06, 0.18 |
| Sex[b] | −0.08 | 0.07 | −0.19, 0.03 | −0.22, 0.05 |
| Group North[c] | −0.02 | 0.19 | −0.33, 0.28 | −0.41, 0.34 |
| Group South[c] | −0.04 | 0.12 | −0.23, 0.15 | −0.26, 0.19 |

Reference categories for the variables presented in this table are as follow:

(a): *Full hand grip*

(b): Female

(c): Group East

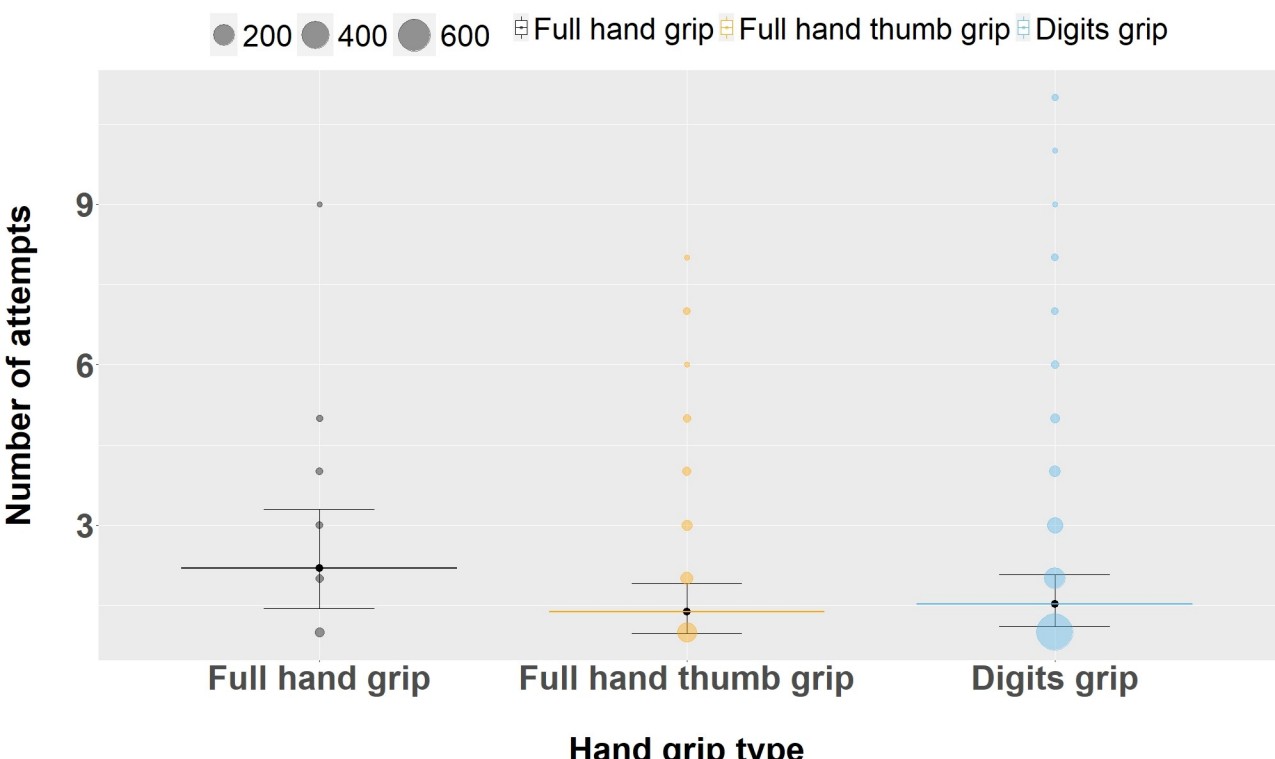

**Fig 3. Effects of the different hand grip types used on the number of attempts to insert stick tools into high-nutrient food sources holes during stick tool use events.** Shown are medians (long thin horizontal lines, *full hand grip* = black, *full hand thumb grip* = orange, *digits grip* = blue) and its 95% CI (error bars). Circle sizes represent the number of events contributing to the number of attempts per hand grip in the dataset. The data and code needed to generate this figure can be found in S1 Data and S1 Code.

accuracy (Model 2). We found an effect of hand grip type on task accuracy. Using the *full hand grip* as the reference level, the model showed that when using either *full hand thumb grip* (estimate: −0.46, CI 95%: [−0.75, −0.16]) or *digits grip* (estimate: −0.36, CI 95%: [−0.63, −0.07]), fewer attempts were needed to access the resource (Table 5). Specifically, the number of attempts to successfully insert a stick decreased by 37% when using *full hand thumb grip* (median: 1.38 ± 95% CI: [0.977, 1.91]) and 31% when using the *digits grip* (median: 1.52 ± 95% CI: [1.105, 2.08]) compared to *full hand grip* (median: 2.19 ± 95% CI: [1.445, 3.3]) (see Fig 3). No substantial effect of age on the accuracy measure was observed (age: estimate: −0.10, CI 95%: [−0.25, 0.06]; Age$^2$: estimate: 0.05, CI 95%: [−0.06, 0.18]), suggesting that all aged individuals were more precise when using the *full hand thumb* or *digits grip*.

Moreover, no difference was observed between *full hand thumb grip* and *digits grip* (estimate: 0.10, CI 95%: [−0.05, 0.25], CI 89%: [−0.02, 0.22]) (see Supporting information S5 Table for full table result of releveled Model 2).

### Fitting specific actions to specific food tasks through ontogeny

We identified 6 stick use actions for extracting food from holes across 8 food tasks. Inspecting if adult chimpanzees would use different hand grips in relation to the specificity of the task they sought to achieve, we contrasted insertion versus pounding actions (Model 3). We found that *digits grip* was the predominant grip for both insertion and pounding (Fig 4), but that there is an increase in the use of the *full hand grip* when performing a pounding action (estimate: 1.38, CI 95%: [0.58, 2.19], the probability of using *full hand grip*: 0.090) compared to the

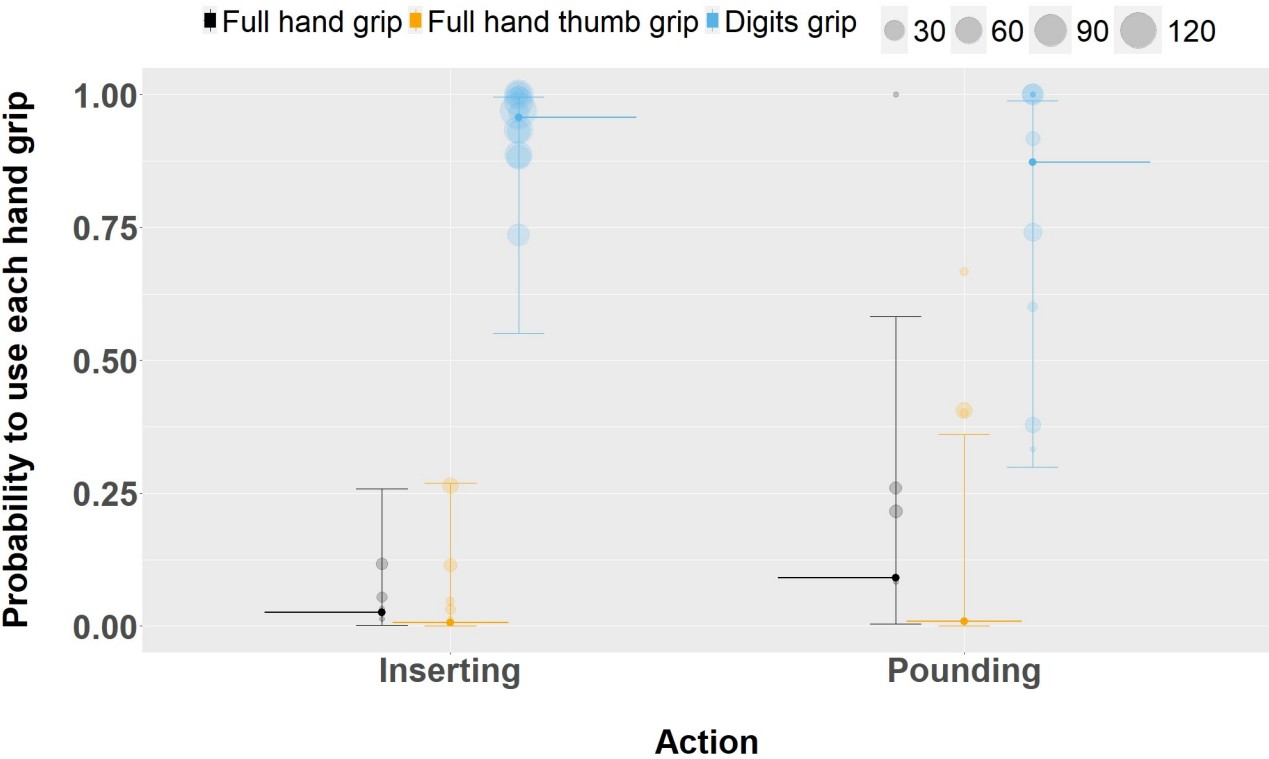

**Fig 4. Effects of different actions on the probability for adult tool users to use the different grip during stick tool use events.** Shown are medians (long thin horizontal lines, *full hand grip* = black, *full hand thumb grip* = orange, *digits grip* = blue) and its 95% credible interval (error bars). Each circle (*full hand grip* = black, *full hand thumb grip* = orange, *digits grip* = blue) represents the sum of occurrences of each hand grip type used per individual in the dataset. The data and code needed to generate this figure can be found in S1 Data and S1 Code.

inserting action (probability to use *full hand grip*: 0.025) (Table 6 (A)). No effect of the action was shown on the probability to use the different hand grip for younger, non-adult, individuals (estimate *full hand grip*: 0.88, CI 89%: [−0.46, 2.23], CI 95%: [−0.78, 2.57]; estimate *full hand thumb grip*: −0.36, CI 89%: [−1.64, 0.94], CI 95%: [−1.93, 1.22]–Table 6 (B)).

Examining if the use of different actions would influence the probability to obtain more food in larvae extraction context (Model 4), we found that there was a higher probability to obtain whole intact larvae when using the levering action (median: 0.47, ±CI 95%: [0.036, 0.88]–Fig 5) compared to any other action (median: 0.11, ±CI 95%: [0.006, 0.50]–Fig 5) in this context (estimate: 2.40, CI 95%: [1.21, 3.72], Table 7). Specifically, in nearly all instances (97%) an intact larvae was obtained when the individuals used the levering action, while the lack of levering use largely led (67%) to unsuccessful larvae extraction (Table 8). The probability of obtaining more food in larvae extraction was not strongly impacted by age (estimate partial: 0.33, CI 95%: [−1.56, 2.14]; estimate whole: −0.17, CI 95%: [−1.72, 1.24], Table 7).

We further tested if the probability to use the levering action in larvae extraction varied through ontogeny (Model 5). We found that non-adult individuals (median: 0.37, ±CI 95%: [0.04, 0.78]–Fig 6) were less likely to use the levering action (estimate: −1.37, CI 89%: [−2.43, −0.17], Table 9) compared to adult individuals (median: 0.70, ±CI 95%: [0.11, 0.94]–Fig 6).

Looking specifically at the levering action in the 2 contexts where it was most represented—larvae extraction (*n* = 66) and nut kernel extraction (*n* = 60), the model showed a moderate effect of age on the use of the levering action (Model 6). The probability to use the levering action increased with age with increases during both larvae extraction (estimate: 0.87, CI 87%:

**Table 6. The effect of action, sex, and group identity on the hand grip type used by adult (model 3A) and non-adult (model 3B) individuals during different tasks of stick tool use events.** Credible intervals of 89% and 95% are presented. Values in bold represent credible intervals excluding zero.

| | Coded level | Term | Estimate | SE | 89% CI | 95% CI |
|---|---|---|---|---|---|---|
| **Model 3 (A) Adult individuals** | *Full hand grip*[a] | Intercept | −3.70 | 1.39 | −5.96, −1.63 | −6.67, −1.00 |
| | | **Pounding**[b] | **1.38** | **0.41** | **0.73, 2.04** | **0.58, 2.19** |
| | | Sex[c] | −1.34 | 0.89 | −2.68, 0.15 | −3.00, 0.52 |
| | | Group South[d] | −0.41 | 0.86 | −1.72, 0.99 | −2.04, 1.35 |
| | *Full hand thumb grip*[a] | Intercept | −5.13 | 2.21 | −8.96, −1.87 | −9.91, −0.96 |
| | | Pounding[b] | 0.52 | 0.37 | −0.07, 1.11 | −0.21, 1.25 |
| | | Sex[c] | 0.50 | 0.90 | −0.98, 1.87 | −1.35, 2.17 |
| | | Group South[d] | −0.59 | 0.97 | −2.11, 0.97 | −2.44, 1.37 |
| **Model 3 (B) Non-adult individuals** | *Full Hand Grip*[a] | Intercept | −3.17 | 2.23 | −7.04, 0.13 | −8.11, 0.80 |
| | | Pounding[b] | 0.88 | 0.84 | −0.46, 2.23 | −0.78, 2.57 |
| | | Sex[c] | 0.03 | 0.97 | −1.51, 1.56 | −1.84, 1.87 |
| | | Group North[d] | 0.01 | 1.00 | −1.58, 1.62 | −1.94, 1.97 |
| | | Group South[d] | −0.40 | 0.98 | −1.96, 1.16 | −2.34, 1.48 |
| | *Full Hand Thumb Grip*[a] | Intercept | −0.52 | 2.30 | −4.29, 2.96 | −5.34, 3.80 |
| | | Pounding[b] | −0.36 | 0.81 | −1.64, 0.94 | −1.93, 1.22 |
| | | Sex[c] | −0.16 | 1.01 | −1.77, 1.42 | −2.13, 1.77 |
| | | Group North[d] | 0.17 | 0.99 | −1.47, 1.75 | −1.78, 2.11 |
| | | Group South[d] | −0.14 | 0.97 | −1.70, 1.41 | −2.09, 1.73 |

Reference categories for the variables presented in this table are as follow:

(a): *Digits grip*

(b): Inserting

(c): Female

(d): Group East

[0.01, 1.78], S6 Table and Fig 7) and nut kernel extraction (estimate: 1.59, CI 85%: [0.02, 3.40], S7 Table and Fig 7).

The probability of using the levering action in larvae extraction increased from 32% (response probability shown in Fig 7: 0.32) at 4 years old (model prediction: 4.19) to above 50% (response probability shown in Fig 7: 0.50) of being used at 10 years old (model prediction: 9.98). This probability continues to increase and reaches 79% (response probability shown in Fig 7: 0.787) probability of being used at 19 years old (model prediction: 19.69). As for the nut kernel extraction context, the probability of using the levering action was always above 50% (response probability shown in Fig 7 at age 2.98: 0.62; response probability shown in Fig 7 at age 18.6: 0.99).

## Discussion

Wild chimpanzees showed protracted development of their hand grip when manipulating a stick as a tool to extract difficult-to-access foods from cavities. While the *digits grip* became the predominant hand grip used by 5 to 6 years old (weaning age), it only became ubiquitous at 15 years old (adulthood). In addition, we found that chimpanzees used 6 actions with stick tools to retrieve 8 food types. Adults, but not non-adults, modulated their behavior to use a more appropriate hand grip to fit action requirements (i.e., power or precision). Such behavioral modulation only by adults indicates an improved understanding of tool use problems and their solutions. Finally, we also found highly protracted development in the use of suitable

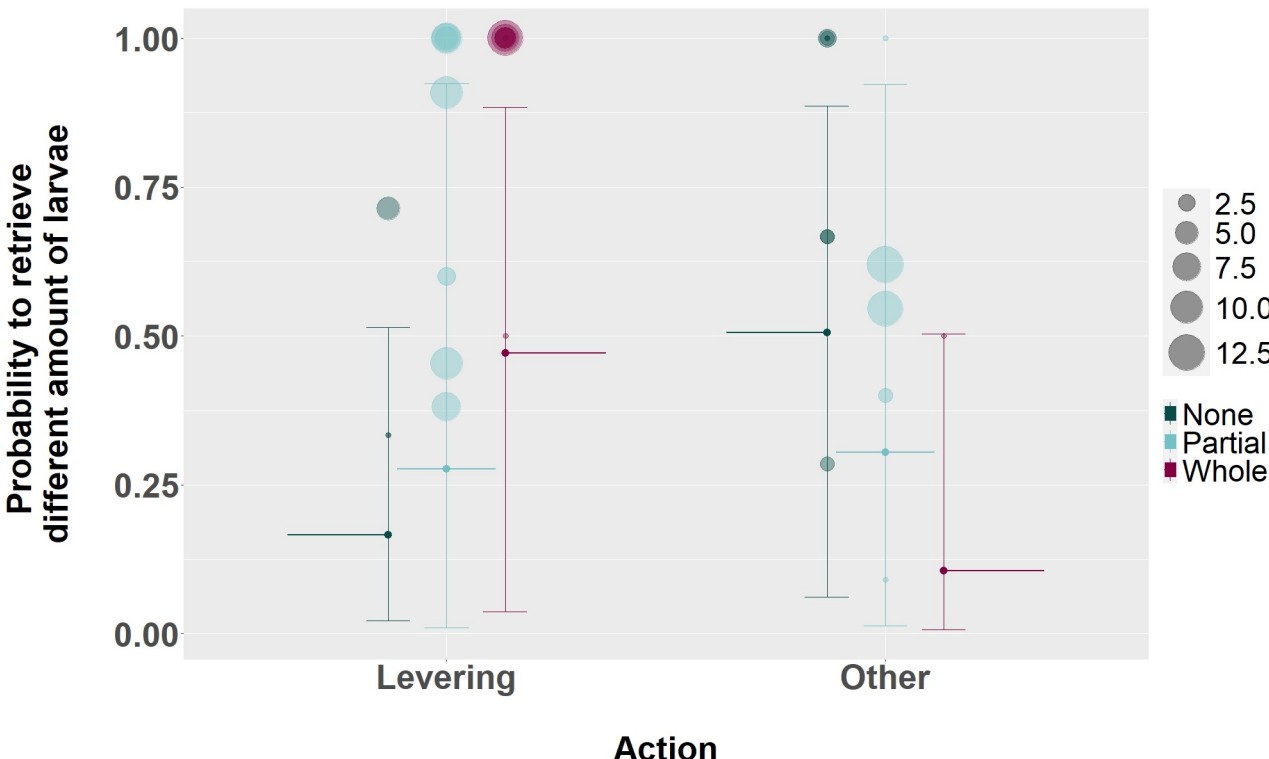

**Fig 5. Effect of the use of the levering action on the probability to retrieve different amounts of larvae with a stick tool in the larvae extraction context.** Shown are medians (long thin horizontal lines, none = green, partial = light blue, whole = magenta) and its 95% credible interval (error bars). Each circle (none = green, partial = light blue, whole = magenta) represents the sum of occurrences of each amount of larvae retrieved per individual in the dataset. S1 Data and S1 Code contain the data and code to generate this figure.

actions to retrieve hidden rather than visible foods, specifically choosing levering in hidden larvae extraction years later than for visible nut kernel extraction. The observed patterns indicate prolonged learning periods are necessary to become efficient in some tool use tasks.

**Table 7. The effect of age class, the use of the levering action and the sex of the individual on the probability to obtain different amounts of larvae with a stick tool during the larvae extraction context.** Credible intervals of 89% and 95% are presented. Values in bold represent credible intervals excluding zero.

|  | Term | Estimate | SE | 89% CI | 95% CI |
|---|---|---|---|---|---|
| **Partial**[a] | Intercept | −0.58 | 1.69 | −3.27, 2.09 | −4.01, 2.70 |
|  | Age class[b] | 0.33 | 0.94 | −1.17, 1.83 | −1.56, 2.14 |
|  | Levering[c] | 0.92 | 0.59 | −0.03, 1.88 | −0.23, 2.07 |
|  | Sex[d] | −0.35 | 0.93 | −1.86, 1.14 | −2.17, 1.44 |
| **Whole**[a] | Intercept | −1.51 | 1.04 | −3.18, 0.13 | −3.63, 0.47 |
|  | Age class[b] | −0.17 | 0.74 | −1.39, 0.99 | −1.72, 1.24 |
|  | **Levering**[c] | **2.40** | **0.64** | **1.41, 3.46** | **1.21, 3.72** |
|  | Sex[d] | 0.44 | 0.79 | −0.85, 1.66 | −1.16, 1.90 |

Reference categories for the variables presented in this table are as follow:

(a): None

(b): Adult

(c): Other

(d): Female

**Table 8. Efficiency outcome of larvae extraction when using and not using the levering action during stick tool use events.**

|  | None | Partial | Whole | Total |
|---|---|---|---|---|
| **Levering** | 7<br>33% | 53<br>65% | 33<br>97% | 93<br>68% |
| **Other** | 14<br>67% | 29<br>35% | 1<br>3% | 44<br>32% |

Legend: Numbers show the number of times the different amount of food was retrieved using and not using the levering action. Percentages show the percentages of times the different amount of food was retrieved using and not using the levering action.

### Chimpanzee hand grips develop slowly through infancy

While all the different categories of hand grip used during stick tool manipulation were observed from infancy to adulthood (see Table 3), modeling the developmental trajectory of chimpanzee stick tool hand grips reveals substantial ontogenetic changes. We observed a succession of changes in the most used hand grip between the ages of 1 and 6 years, shifting from *full hand grip* (no independent digits used to guide the stick) to *full hand thumb grip* (1 independent digit used to guide the stick) to *digits grip* (2 or 3 independent digits used to guide the stick).

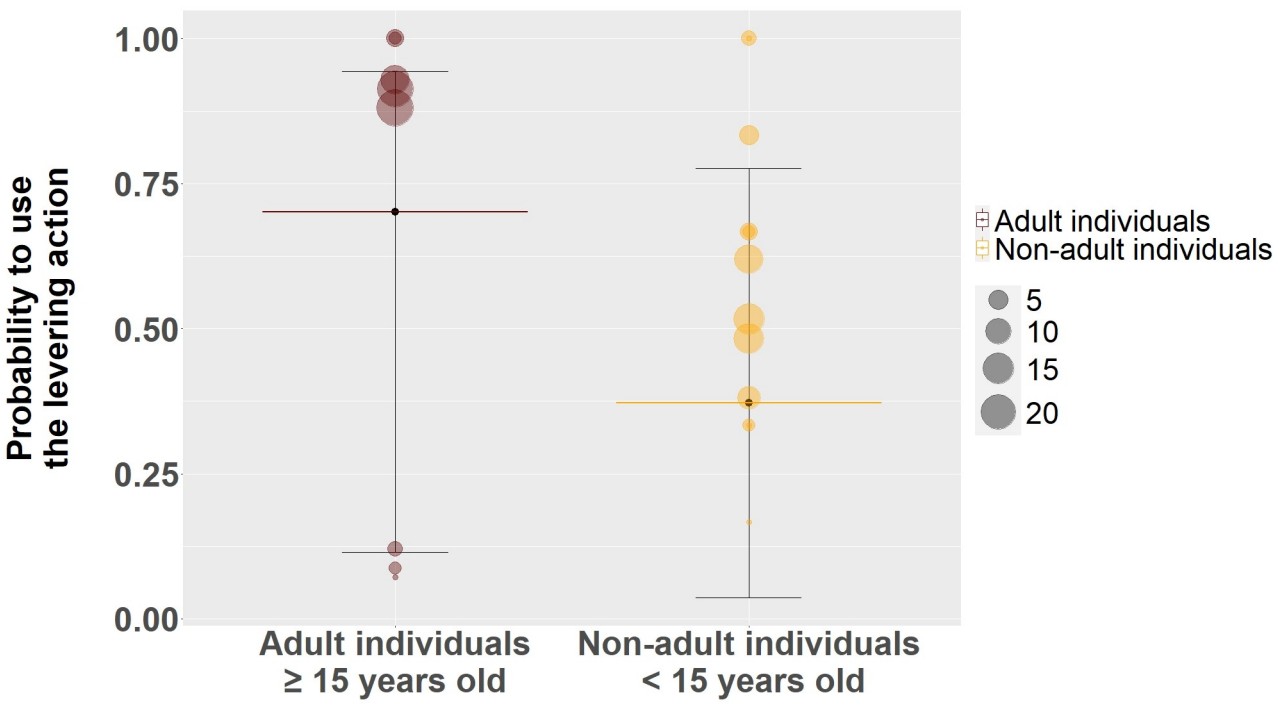

**Fig 6. Effect of the age-class on the probability to use the levering action with a stick tool in the larvae extraction context.** Shown are medians (long thin horizontal lines, Adult individuals = brown, Non-adult individuals = yellow) and its 95% credible interval (error bars). Each circle (Adult individuals = brown, Non-adult individuals = yellow) represents the sum of occurrences of each levering action performed per individual in the dataset. S1 Data and S1 Code contain the data and code to generate this figure.

**Table 9. The effect of age class and the sex of the individual on the probability to use the levering action with a stick tool during larvae extraction context.** Credible intervals of 89% and 95% are presented. Values in bold represent credible intervals excluding zero.

| Term | Estimate | SE | 89% CI | 95% CI |
|---|---|---|---|---|
| Intercept | 0.72 | 1.24 | −1.39, 2.44 | −2.05, 2.81 |
| Age class[a] | −1.37 | 0.71 | **−2.43, −0.17** | −2.68, 0.16 |
| Sex[b] | 0.47 | 0.72 | −0.72, 1.56 | −0.98, 1.82 |

Reference categories for the variables presented in this table are as follow:

(a): Adult

(b): Female

The age by which there was predominant use of the *digits grip* was similar to that demonstrated for fine motor control of objects [72,73]. Independent use of multiple digits during "in-hand" object manipulation, where objects are moved in one hand via manipulation of the digits, also becomes fine-tuned around the age of 5 years [74]. Adult-like physical manipulation of objects is likewise shown to develop by weaning age across various primate species [73]. This indicates a general primate pattern of neuromotor maturation of fine motor control by weaning age, and likely also explains the developmental trajectory of the hand grip use to

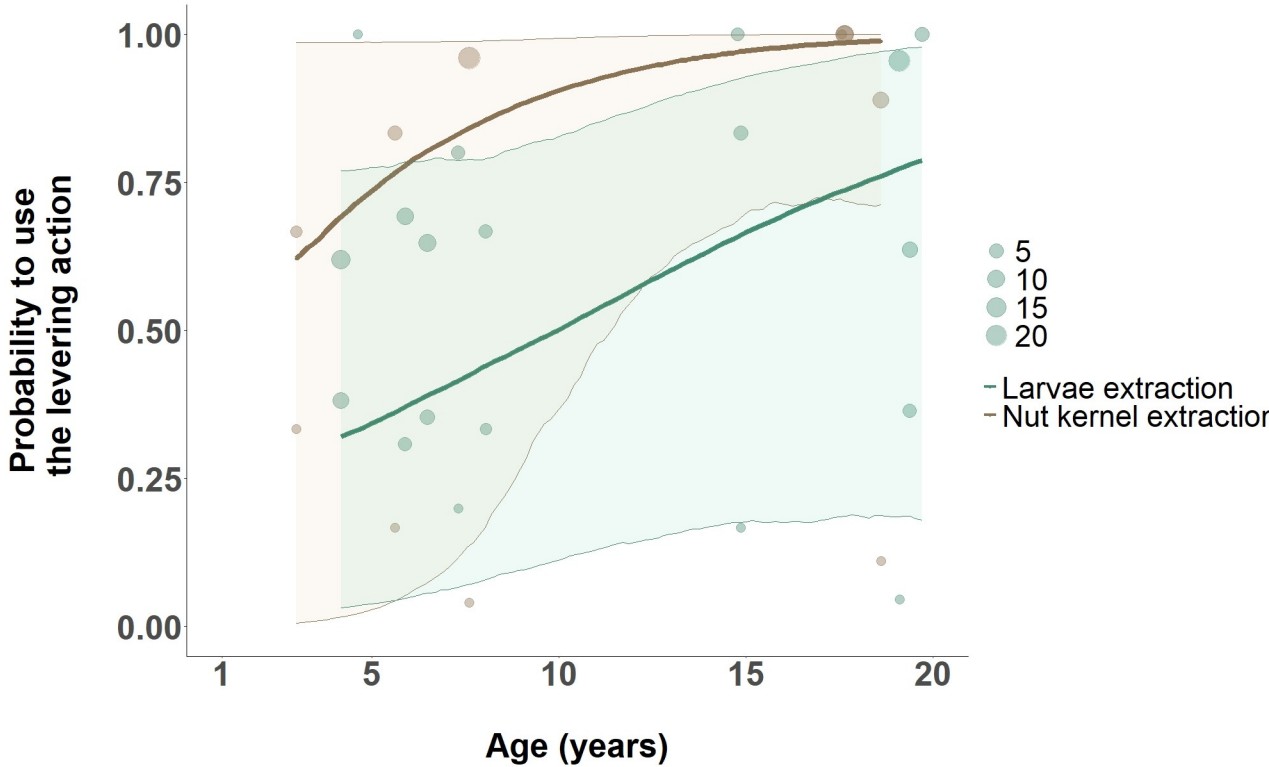

**Fig 7. Age effect on the probability of using the levering action with a stick tool in larvae extraction and nut kernel extraction.** The regression lines obtained from the Bayesian regression model (nut kernel extraction = brown, larvae extraction = green) with a 95% credible interval are shown. Each circle (larvae extraction = green, nut kernel extraction = brown) represents the sum of occurrences of levering action performed per individual in the dataset. S1 Data and S1 Code contain the data and code to generate this figure.

manipulate stick tools observed in chimpanzees. However, neuromotor maturation is less likely to explain further changes in the usage of *digits grip* between the ages of 5 and 15 years.

## Chimpanzee hand morphology is suitable for grips allowing both power and precision

In examining whether different hand grips improve the accuracy of inserting a stick into a cavity, we showed that both the *full hand thumb grip* and *digits grip* were similarly accurate and better than the *full hand grip*. Yet, adult chimpanzees ubiquitously use the *digits grip*, potentially due to the specific characteristics it affords.

Compared to the hand grip found in humans, the *digits grip* would be classified as an "intermediate" grip [61], where "elements of power and precision are present roughly in the same proportion" [75]. Both the *full hand grip* and *full hand thumb grip*, however, would be classified as power grips (i.e., a grip typically described as optimal for applying force, [23,61]). The 3 categories of *digits grip* found in Taï chimpanzees (see Fig 1) are analogous to (i) the lateral grip (*digits grip* category 1) used for example, for keys; (ii) the adduction grip used for example, to hold a cigarette (*digits grip* category 2); and (iii) the tripod variation grip used for example, to manipulate a scalpel (*digits grip* category 3) in humans [61,75].

Chimpanzees are proficient tool users, and their hand morphology represents a compromise between manipulation and locomotion [23,24]. Their thumb is relatively short compared to the length of the other digits [76], thought to create some anatomical constraints that possibly hinder the maneuverability of objects and tools [22]. While chimpanzees can oppose their thumb to their other digits [77], the skeletal architecture of their wrist and palm bones limits their ability to cup their palm [78], which in humans likely contributes to precision handling. Their short thumb offers less possibility to apply force when compared to humans, a difference in torque between the 2 species that has also been argued to explain the difference in precise handling of objects or tools [79].

An intermediate grip might therefore be the most efficient stick tool use option with respect to chimpanzee's hand morphology, as it allows for precise manipulation but also offers the flexibility to apply force without relying extensively on the thumb. In addition, while we observed that the *full hand thumb grip* is as accurate as the *digits grip* in the inserting task, its principal attribute lies in its ability to apply force, which might make it less efficient for other actions, such as levering, compared to the *digits grip*.

## Adults modulate hand grip to suit the required action more than immature individuals

In our non-exhaustive dataset, chimpanzees in Taï use sticks to execute at least 6 different actions in 8 different food extraction tasks (Table 1—see also Boesch and Boesch, 1990 [4]). Adults (but not non-adults) could modulate their hand grip depending on the action required. When power was needed (pounding action), adult chimpanzees chose more often the *full hand grip* (i.e., a power grip). When precision was needed (inserting action), the *digits grip* (i.e., an intermediate grip) was preferred. Such behavioral modulation in adulthood suggests a better understanding of task requirements later in ontogeny. These results are similar to the shift from a power grip in the perforation task (i.e., perforating surface to access ground nest) to an intermediate grip in a fishing task while trying to access termites observed in Goualougo adult chimpanzees [45]. The ability of adult chimpanzees to choose the right grip for the right action, something not observed in all tasks in non-adults, suggests that the necessary cognitive flexibility to understand the task and apply the right action is reached late in ontogeny.

## Prolonged acquisition of action to task matching

On top of using the right grip for the right action, the ability to use the correct action to extract different food sources was also acquired across ontogeny. Specifically, the extraction of a hidden food (i.e., larvae) required the longest to master the action preferred by adults (i.e., levering). Levering, the most efficient action for retrieving intact larvae, was more consistently used by adult compared to non-adult individuals. It became the predominant action for this food only at 10 years old, but its probability of being used further increases into adulthood, suggesting a better understanding of the task requirements later in ontogeny. Younger individuals rather used inadequate actions (such as screwing or pounding) which resulted in mashing the larvae in the hole, enabling retrieval of only larvae parts.

However, the prolonged learning period was not related to the action of levering itself, as in a context with visible food (i.e., nut kernel extraction), levering was already predominantly used before 5 years old (Fig 7). We suggest rather that the slow acquisition to use levering to extract larvae is related to the additional cognitive demands of extracting a hidden and awkwardly placed resource. A single larvae sits at the end of a narrow hole several centimeters in length bored into a tree trunk with a narrow entrance. The larvae fills the breadth of the hole but cannot be seen when in the hole. Extracting the larvae intact requires delicate maneuvering of the stick under the larvae to lift it towards the hole entrance (i.e., levering). In contrast, for nut kernel extraction, the nutshell offers a large opening, aiding tool maneuvering and kernel visibility.

We argue that efficient extraction of the intact larvae from a hole, which could be referred to as a haptic task since it must rely only on sense of touch not sight, likely requires cognitive control. Cognitive control implies working towards a long-term goal, ignoring irrelevant information, and controlling impulses to accurately understand and predict task requirements, especially when food sources are hidden from view [80]. Cognitive control in this context likely contributes to the decision-making of generating the right action to efficiently extract the food source.

## Evolutionary perspective—Hand morphology

While morphological features might hinder tool manipulation, some species find alternative strategies for tool use, although using few tools per species. Tufted capuchins (*Sapajus apella*) have a hand morphology that features a saddle-shaped trapeziometacarpal but currently there is no agreement as to how much it allows the species to achieve thumb opponency [20,81,82]. Nonetheless capuchins can achieve nut cracking using standing posture and bimanual grasp [83,84]. New Caledonian crows (*Corvus moneduloides*)—and to a lesser extent Hawaiian crows (*Corvus hawaiiensis*)—likely achieve tool use skills through adapted bill morphology [18,19]. Wild and captive orangutans use their mouth and teeth to manufacture and use tools in extractive contexts [85–87]. It is feasible that the hand grip options available to chimpanzees allow for their expanded tool kit compared with other nonhuman species.

Hand morphology and hand grips have been suggested to play a key role in the evolution of hominin tool technologies [21–24]. One hypothesis in particular posits that early hominin manual anatomy (considered analogous to that of apes) may have prevented an earlier emergence of flaked stone tool technologies, as the forceful precision grips required for flaked tool use may have been ineffectively performed by these species [22]. However, recent studies on our closest living relatives, chimpanzees and bonobos, have shown their ability to use one of the forceful precision grips, the lateral grip [61] in feeding contexts (wild chimpanzees: [63]) and when using flaked stone tools (captive bonobos: [88]). As the lateral grip is among the most frequently recruited grips by modern humans when using flaked stone tools [22,89],

Cebeiro and Key [88] argued that the *Pan*-like (and therefore potentially early hominin) anatomy was capable of effectively securing flaked stone tools.

Chimpanzees in our study also used the lateral grip (i.e., *digits grip* Category 1) to hold and manipulate stick tools to extract food resources, and do so to apply both precision and power according to the task requirements (Table 1). Our results therefore add to previous findings by showing chimpanzee's ability to use grips argued to be essential in flake stone tool use and production, suggesting that the hand morphology of ancient hominins might not have been the main constraint on the emergence of earliest flaked stone tool technologies (although see [90] —different stone tool technologies have different dexterity, cognition, and coordination requirements).

## Evolutionary perspective—Skill acquisition and the prolonged juvenile dependency hypothesis

We evaluated the ability to use suitable actions in contexts of variable complexity based on their problem-solution distance [14,15]. Relative to nut kernel extraction where the target is directly visible, larvae extraction involves a greater problem-solution distance as the target is embedded within a substrate and is therefore not visible. As such, we argue that larvae extraction is likely a more cognitively demanding task than nut kernel extraction.

Wild studies have demonstrated cognitive skills used by primates in tool use tasks, showcasing their ability to learn to choose tools with the right property but also their ability to combine or use tools in sequence [91–95]. Likewise, cultural transmission experiments were carried out in captivity in which chimpanzees demonstrate that they socially learn to use different forms of tool use to solve foraging challenges [96,97].

We observed a slower developmental progression in choosing suitable actions for larvae extraction, extending well beyond the age when stick motor control was mastered. In this context, protracted learning of action-to-task mapping continued into teenage years. This pattern suggests that task learning is essential and additional to motor control for efficient stick tool use behavior, and that the accumulation of skills required for successful food extraction can take many years, especially for more challenging tasks. From a cognitive perspective, our data do not permit us to comment on which aspects of cognitive acquisition explain this ontogenetic pattern, but relevant candidates highlighted by previous research include but are not limited to reasoning [13,26], conceptual knowledge about objects [27], or working memory [14].

Interestingly, chimpanzees continue to travel predominantly in their maternal unit until 12 years old [98], giving offspring substantial opportunity to continue to learn from their mothers until close to adulthood. How much of the task learning occurs through social rather than by trial-and-error learning remains to be tested. Several studies demonstrate that social facilitation influences chimpanzee learning of object manipulation tasks in captivity [99,100]. In the wild, a prolonged developmental period and maternal dependency are suggested to facilitate the gradual learning of complex foraging tasks ([42,55,57,101,102,103]; although see [104,105]). Anecdotal observations demonstrate that chimpanzees continue to closely observe others (peering) after their juvenile years, especially when unusual objects are being manipulated or unusual actions are performed with them [106–108].

The pattern of protracted stick tool use learning fits with the prolonged juvenile dependency hypothesis [17], that rapid encephalization seen in the hominoid lineage, may reflect the capacity to continue learning new skills that increase foraging capacities at least into adulthood. Whether chimpanzees learn such tasks more from mothers than other community members remains to be tested.

## Conclusions

Our results are in line with Hunt and colleagues' [12] hypothesis, showing that the development of complex manipulation in chimpanzees is not only dependent on morphological features or neuro-motor maturation but also has a considerable cognitive component essential for flexible tool use expression. Learning and understanding the action needed to efficiently extract food plays a significant role in developing adequate adult-like skills which may well be facilitated by the protracted dependency period experienced by wild chimpanzees. Motor and cognitive maturation likely work in concert, in informing the development of hand grip to use stick tools during ontogeny.

Wild chimpanzees seem to be able to find alternative strategies to limit their hand anatomy restrictions by choosing hand grips that allow them to apply both power and precision when using stick tools. Ontogenetic analyses demonstrate that morphological features are likely not the principal limiting factor in allowing the expression of flexible tool use behavior, as more cognitively challenging tasks were slower to develop. Task experience and decision-making seem essential for stick tool use behavior to occur across a range of contexts. Chimpanzees have one of the most extensive tool kit of nonhuman animals [2]. Whether the limiting factor preventing more tool use and manufacture in chimpanzees populations is cognitive capacity in specific domains (e.g., action planning) or is rather due to limited exposure to role models to acquire culture cumulatively (see [109,110]) remains to be examined.

## Supporting information

**S1 Table. Number of observations of hand grip used to hold stick tools recorded in the different contexts and age-class.**
(DOCX)

**S2 Table. Number of observations of tasks operated with stick tools recorded in the different age-class.**
(DOCX)

**S3 Table. Number of observations of actions operated with stick tools in the different context recorded in the different age-class.**
(DOCX)

**S4 Table. Number of observations of stick use events and number of videos associated per individuals in the dataset.**
(DOCX)

**S5 Table. The effect of the hand grip type used, the age of the individual performing the task, its sex, and its group identity on the number of attempts to insert stick tools into high nutrients food holes (*Full hand thumb grip*[a], Female[b], and Group East[c] as reference categories).** Credible intervals of 89% and 95% are presented. All numeric predictor variables were standardized to mean = 0 and sd = 1. Values in bold represent credible intervals excluding zero.
(DOCX)

**S6 Table. Bayesian regression model results of the effect of age on the probability of using the levering action in the larvae extraction context (Model 6A). Credible intervals of 87%, 89%, and 95% are presented.** Values in bold represent credible intervals excluding zero.
(DOCX)

**S7 Table. Bayesian regression model results of the effect of age on the probability of using the levering action in the nut kernel extraction context (Model 6B). Credible intervals of**

**85%, 89%, and 95% are presented.** Values in bold represent credible intervals excluding zero. (DOCX)

**S8 Table. Non-exhaustive table of the terminology used in the literature to describe different grips.**
(DOCX)

**S1 Data. Table containing all relevant data to replicate this study.**
(CSV)

**S1 Code. Script to be used to replicate the results and figures presented in this study.**
(R)

## Acknowledgments

We would like to thank the Ministère de l'Enseignement Supérieur et de la Recherche Scientifique, the Ministère des Eaux et Fôrets in Côte d'Ivoire and the Office Ivoirien des Parcs et Réserves for allowing us to carry out this study. We are very grateful to Frans de Waal who supported the publication of this study as Academic Editor until his recent passing. This research is dedicated to the late Christophe Boesch for creating and nurturing the Taï Chimpanzee Project since 1979. We thank the Centre Suisse de Recherches Scientifiques en Côte d'Ivoire for their logistical support. We also thank Valeria Ferrario, Christelle Nihouarn, and Ann-Sophie Warkentin for helping with the inter-reliability test. We thank Aisha Bründl, Jenny Jaffe, and Christelle Nihouarn for their assistance in collecting some of the video data, to Oscar Nodé-Langlois for his chimpanzee hand drawings and to the staff of the Taï Chimpanzee Project for their support. A special thank you to Tracy Kivell, Tina Petersen, and Derry Taylor for their valuable comments on the manuscript.

## Author Contributions

**Conceptualization:** Mathieu Malherbe, Catherine Crockford, Roman M. Wittig.

**Data curation:** Mathieu Malherbe, Liran Samuni, Roman M. Wittig.

**Formal analysis:** Mathieu Malherbe, Liran Samuni.

**Funding acquisition:** Catherine Crockford, Roman M. Wittig.

**Methodology:** Mathieu Malherbe, Sonja J. Ebel, Kathrin S. Kopp.

**Supervision:** Liran Samuni, Catherine Crockford, Roman M. Wittig.

**Writing – original draft:** Mathieu Malherbe, Liran Samuni, Catherine Crockford, Roman M. Wittig.

**Writing – review & editing:** Mathieu Malherbe, Liran Samuni, Sonja J. Ebel, Kathrin S. Kopp, Catherine Crockford, Roman M. Wittig.

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
