## [Editor Report · Decision Letter 0]

26 Nov 2023

Dear Dr Malherbe, 

Thank you for submitting your manuscript entitled "Prolonged development of stick tool use skills in wild western chimpanzees" for consideration as a Research Article by PLOS Biology.

Your manuscript has now been evaluated by the PLOS Biology editorial staff, as well as by an academic editor with relevant expertise, and I'm writing to let you know that we would like to send your submission out for external peer review.

Once your full submission is complete, your paper will undergo a series of checks in preparation for peer review. After your manuscript has passed the checks it will be sent out for review. To provide the metadata for your submission, please Login to Editorial Manager (https://www.editorialmanager.com/pbiology) within two working days, i.e. by Nov 28 2023 11:59PM.

Kind regards,

Roli Roberts

Roland Roberts, PhD

Senior Editor

PLOS Biology

rroberts@plos.org

---

## [Decision Letter · Decision Letter 1]

17 Jan 2024

Dear Dr Malherbe,

Thank you for your patience while your manuscript "Prolonged development of stick tool use skills in wild western chimpanzees" went through peer-review at PLOS Biology. Your manuscript has now been evaluated by the PLOS Biology editors, an Academic Editor with relevant expertise, and by three independent reviewers.

You'll see that reviewer #1 is very positive, and only has minor textual requests. Reviewer #2 is impressed by the quality of the study, but wants you to improve your discussion of the existing literature, reduce your reliance on corvid work, and tone down your “unconvincing” assumptions about cognition. However, all of his/her requests are textual in nature. Reviewer #3 only has a few minor requests, but is uncertain whether there is enough novelty. (Note that because of our confusing file nomenclature, this reviewers mistakenly thinks that this was a revised manuscript).

IMPORTANT: I discussed these comments with the Academic Editor, who shares some of these concerns, but agrees that you should be given an opportunity to address them. The Academic Editor also had advice about the framing and Title. Specifically, the Academic Editor said:

"I agree with reviewers that the cognitive component is not worked out well in the paper. The statements about cognition sound like hand waving. I also agree with reviewer #2 that they underestimate how many animals use tools. For example, is the building of beaver dams not complex tool use? It depends on the definition. Capuchin monkeys seem almost as complex in their tool use among the primates as chimpanzees. I still think the most striking finding is the slowness of acquiring skills. I think the paper should emphasize that species with such complex skills need a lengthy development, like apes, elephants, and whales, to acquire all the skills they need in life. Most readers won't realize that apes are about as slow developers as humans, being adult only by 16 yrs. I'd emphasize this, also in the title."

In light of the reviews, which you will find at the end of this email, we are pleased to offer you the opportunity to address the comments from the reviewers in a revision that we anticipate should not take you very long. We will then assess your revised manuscript and your response to the reviewers' comments with our Academic Editor aiming to avoid further rounds of peer-review, although might need to consult with the reviewers, depending on the nature of the revisions.

**IMPORTANT - SUBMITTING YOUR REVISION**

*Resubmission Checklist*

*Published Peer Review*

*PLOS Data Policy*

*Blot and Gel Data Policy*

Sincerely,

Roli Roberts

Roland Roberts, PhD

Senior Editor

PLOS Biology

rroberts@plos.org

REVIEWERS' COMMENTS:

Reviewer #1:

This manuscript presents a high-quality and meticulously conducted and analysed study of the details of manual manipulation of tools through the lives of wild chimpanzees. It is clearly presented and sets the work in the context of a thorough review of the most relevant research literatures, including those concerning comparisons with humans and hominin evolution. 

The analysis distinguishes the principal kinds of tool grips and their occurrence through ontogeny, demonstrating that the most finely skilled aspects of tool handling are not maturing until adulthood. This is a striking discovery, appropriate to this journal. I found figures 2 and 7, nicely portraying the numerical analyses of these changes over time, very clear and helpful. The findings will be of most immediate interest to others studying the anatomy, functional use and cognitive requirements of tool use in humans, earlier hominins and non-human primates, but also relevant to tool use, learning and culture more generally across humans and other animal taxa.

I have only a few more specific comments, amounting to suggestions for minor revision - by line number: - 

68 - a reader may expect a reference here to the Rutz et al paper cited, on Hawaiian crows.

Also note (and line 91) the recent discoveries of Goffins cockatoos using a tool set of three different tools - although this has been found in only a minority of their population studied: 

O'Hara, M. et al. Wild Goffin's cockatoos flexibly manufacture and use tool sets. Curr Biol 2023.

113 - precision grip - here I would think it may be expected to comment on the significant aspects of anatomy that are so different, which we do not read about until the Discussion.

Introduction - Mention of cultural learning is noticeably absent from the Introduction although references such as those by Biro, Musgrave and Whiten are focused on this. The Discussion picks up on this issue but I feel it ought to get at least a mention in the Introduction too.

175 - How many inter-observer tests were done?

275 and 287 - Why the choice of 'age squared' rather than other functions?

707 - What is the nature of the evidence for social learning in these studies, in the wild? In captivity there is experimental evidence showing chimpanzees will acquire different forms of tool use to solve a foraging challenge through social learning, notably cultural transmission experiments by A Whiten et al (Nature 2005, Curr Biol 2007). The authors might consider citing these as adjuncts that support the circumstantial evidence from the wild.

Minor 'typo level' points

Its not really my job to highlight these but here are a few noted: -

275 - grammar - delete 'A' ?

282 - apostrophe - grips'

300 - 'where we only included adult' better as 'where we included only adult …'

322 - 'confirm' -> 'confirmed' ?

512 - 'which increases' -> 'with increases' ?

553 - 'object' -> ' objects' ?

END

Reviewer #2:

The authors present an unusual manuscript. There are very few studies that offer such a well- documented and detailed presentation of the ontogeny of tool use behavior. From that perspective, this is a refreshing and important addition to the literature. 

However, there are three significant concerns that should be addressed. First, some sections require a much more thorough review of the literature. Second, there is an over-reliance on comparisons with the Corvid literature which must be addressed. Third, the assumptions about "cognition" are frequently vague and unconvincing. There are comments related to those concerns listed below. 

The descriptions and documentation of the ontogeny of the tool behavior are the clear strengths in this paper. It can easily stand confidently on those data alone. The paper would be greatly improved if the assertions about cognition were removed. They are simply too vague and unconvincing, and they weaken the overall effort. In places, they seem to be inaccurate and misinformed. 

I congratulate the authors on an excellent effort in major sections of the paper. In the case of this manuscript, less would certainly be much more in terms of the final product. 

49-50 The authors have cited only partially quoted Beck's (1980) definition of tool use. Also, that definition was revised in Shumaker et al. (2011). It would be preferable more accurate if the authors refer to the full and revised definition. 

51-60 It is not clear how the authors draw the conclusion that tool use is typically differentiated into stereotyped and flexible tool use. While some authors have stated that, it is an overgeneralization as presented here. It would be reasonable to say some authors have suggested these labels. Hunt takes a fairly narrow view informed primarily from corvid tool use. But the idea that there are a "handful of birds and mammals" - presumably referring to species - that flexibly use tools is not well established or firmly documented. The authors should be clearer with this statement. While certainly true that some species demonstrate greater flexibility with tool use and manufacture the specifics are not well established. 

64-81 The statements being made by the authors are not accurate or consistent with the literature on tool use. There are other species that habitually use tools flexibly. This section suggests that a more thorough review of the literature is essential, with a significant revision of the statements being offered. It also suggests that only the chimpanzee and corvid literature has been thoroughly reviewed, and that the literature on New Caledonian crows is being over emphasized and perhaps over interpreted. 

82-108 As with the Introduction, the authors cite sources (primarily Byrne) that are presented as widely accepted regarding the relationship between tool use and cognition. That is not the case, and a more thorough review is required here. It is very fair to introduce these concepts, but that must be done accurately. 

188-220 Very nicely done. The illustrations are very helpful. 

335-338 The authors do a very good job offering detailed descriptions of the behaviors being documented. The suggested correlations to presumed cognitive complexity, or functions, are not well established or explained. These is pervasive throughout the manuscript. 

683-732 The conclusions about chimpanzee cognition based on the Corvid model are tenuous and should be reconsidered. As mentioned in an earlier comment, there is an over reliance on Hunt throughout the manuscript and a broader literature review and overall perspective is essential. 

749-750 This is a vague statement without citation. Chimpanzees and orangutans use all of the same modes of tool use in all settings. It is accurate to say that chimpanzees use more tools in combination with each other than other species except humans. However, the frequency of tool use in chimpanzees, or any other species, has not been documented. As a result, broad generalizations need to be made more cautiously. 

Reviewer #3: 

This study describes the use of various techniques in the use of sticks in wild Western chimpanzees in a long term field site. I found the study well written and compelling, with very little to add or issues for the authors to address apart from the point below. Sadly I could not assess if and how previous comments by other reviewers have been addressed. In addition, I am also not really clear about the novelty brought by these results, which is not conveyed particularly well in the manuscript. What does it teach us beyond what is already in the literature, including in studies produced by several of the co-authors in this article?

L706-710: The construction of this sentence is weird as it suggests that the references can refer to the first part of the sentence (that wild primates can SOCIALLY learn to choose tools with the right property). Yet, two of the cited papers are reviews, and the Sanz et al. paper does not show socially learn processes. To my knowledge, the only papers that could simultaneously sustain both points made by the authors regarding the socially learn use of the proper tools in wild primates would be the moss-sponge case described in wild chimps but even then, the studies would have to be combined to carry the point made (Hobaiter et al, 2014 Plos Bio, for social learning and Lamon et al. 2018 Proc Soc B for the right properties). But I may overlook other empirical studies.

---

## [Decision Letter · Decision Letter 2]

22 Mar 2024

Dear Dr Malherbe,

Thank you for your patience while we considered your revised manuscript "Protracted development of stick tool use skills into adulthood in wild western chimpanzees" for publication as a Research Article at PLOS Biology. This revised version of your manuscript has been evaluated by the PLOS Biology editors, the Academic Editor and one of the original reviewers.

Based on the review , we are likely to accept this manuscript for publication, provided you satisfactorily address the following data and other policy-related requests.

IMPORTANT - please attend to the following minor issues:

a) Your Title currently lacks an active verb. Please can you change it to the following, which we think is more appealing and informative: “Development of stick tool use skills in wild western chimpanzees can take up to 15 years” (or maybe “Development of stick tool use skills is prolonged in wild western chimpanzees,” if that is too specific)

b) Thank you for supplying the raw data and the R script. Can you confirm that these are sufficient to generate all of the Figures? If not, we will also need the numerical values underlying Figs 2, 3, 4, 5, 6, 7, either as a supplementary data file or as a permanent DOI’d deposition.

c) Please cite the location of the data clearly in all relevant main Figure legends, e.g. “The data and code needed to generate this Figure can be found in S1 Data and S1 Code" (you would need to rename "Dataset - PBIOLOGY-D-23-03043R1.csv" to S1_Data.csv" and "script.R" to "S1_Code.R").

We expect to receive your revised manuscript within two weeks. 

*Published Peer Review History*

*Press*

Sincerely,

Roli Roberts

Roland Roberts, PhD

Senior Editor

rroberts@plos.org

PLOS Biology

DATA POLICY:

Regardless of the method selected, please ensure that you provide the individual numerical values that underlie the summary data displayed in the following figure panels as they are essential for readers to assess your analysis and to reproduce it: Figs 2, 3, 4, 5, 6, 7. NOTE: the numerical data provided should include all replicates AND the way in which the plotted mean and errors were derived (it should not present only the mean/average values).

CODE POLICY

Per journal policy, as the code that you have generated is important to support the conclusions of your manuscript, we require that you make it available without restrictions upon publication. Please ensure that the code is sufficiently well documented and reusable, and that your Data Statement in the Editorial Manager submission system accurately describes where your code can be found.

DATA NOT SHOWN?

REVIEWER'S COMMENTS:

Reviewer #2:

The authors have provided a detailed response to all reviewer comments along with a revised manuscript. 

The authors have done an exceptionally good job of acknowledging all comments, editing the manuscript appropriately, and explaining those edits in their reply. I am impressed with their collegiality and overall response to the feedback that was provided. 

As a result, the manuscript is greatly improved and I strongly recommend publication. 

I congratulate the authors on an excellent paper that will make an important contribution to the literature. They would be very proud. 

I also congratulate the editor and other reviewers. I believe this has been a model process for a manuscript review. 

It has been a privilege to have been part of the review. Thank you.

---

## [Editor Report · Decision Letter 3]

28 Mar 2024

Dear Dr Malherbe,

Thank you for the submission of your revised Research Article "Protracted development of stick tool use skills extends into adulthood in wild western chimpanzees" for publication in PLOS Biology. It is somewhat bittersweet to be able to say that on behalf of my colleagues and the Academic Editor, Frans de Waal, who sadly recently passed away, but who remained very supportive of this study to the end, I'm pleased to say that we can in principle accept your manuscript for publication, provided you address any remaining formatting and reporting issues. These will be detailed in an email you should receive within 2-3 business days from our colleagues in the journal operations team; no action is required from you until then. Please note that we will not be able to formally accept your manuscript and schedule it for publication until you have completed any requested changes.

Sincerely, 

Roli Roberts

Senior Editor

PLOS Biology

rroberts@plos.org